# Reliable Compositional Editing with Overlap-Aware Attention in Diffusion Models

**Salamata Konate** *
skonate@yorku.ca

**Hassan Hamidi** †
hhamidi@yorku.ca

**Elham Dolatabadi** *
edolatab@yorku.ca

**Frank Rudzicz** ‡
fr591304@dal.ca

**Laleh Seyyed-Kalantri** *
lsk@yorku.ca

## Abstract

Despite significant advances in diffusion models, achieving precise, composable image editing without task-specific training remains a challenge. Existing approaches often rely on iterative optimization or linear latent operations, which are slow, brittle, and prone to entangling attributes (e.g., lipstick altering skin tone). We introduce SphereEdit, a training-free framework that leverages the hyperspherical geometry of CLIP embeddings to enable interpretable, fine-grained control. We model semantic attributes as unit-norm directions on the sphere and show that it supports clean composition via angular controls. At inference, SphereEdit uses spherical directions to modulate cross-attention producing spatially localized edits across diverse domains without optimization or fine-tuning. Experiments demonstrate sharper, more disentangled adjustments. SphereEdit provides a geometrically grounded, plug-and-play framework for controllable and composable diffusion editing.

## 1 Introduction

Diffusion models have revolutionized image synthesis, creating demand for precise, composable editing that preserves identity while enabling controllable modifications[1–4]. Despite remarkable generative quality, two fundamental challenges persist: limited understanding of how semantic attributes interact across denoising steps and lack of interpretable control over latent space semantics [1, 4–7].

These challenges have motivated various editing approaches, but each faces critical limitations. Training-free optimization techniques are slow and prompt-fragile [8, 9], while linear latent edits leak into unrelated regions and compose poorly [10–12]. Attention manipulation improves semantic control but typically relies on heuristics with weak spatial localization, leading to artifacts and attribute spillover [12, 13]. These approaches collectively prevent clean multi-attribute composition and token-aware control.

To address these two failure modes—compositional inconsistency and spatial spillover —we combine a geometric controller with token-aware spatial attribution. While recent work studies diffusion geometry via Riemannian metrics or isometric mappings that require retraining [7, 14], we propose **SphereEdit**, a training-free alternative that exploits the natural hyperspherical structure of CLIP embeddings. We model each attribute as a unit-norm direction on the sphere and compose multiple attributes with simple angular weights, yielding bounded, interpretable edits. In parallel, cross-attention attribution provides token-aware soft maps that localize where each attribute should act

---

*York University and Vector Institute, † York University, ‡ Dalhousie University and Vector Institute

39th Conference on Neural Information Processing Systems (NeurIPS 2025) Workshop: Reliable ML.

**(a) Vanilla diffusion (one step):**

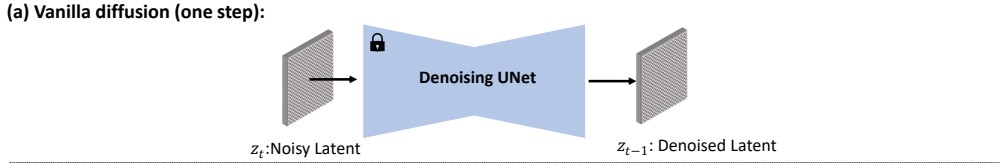

$z_t$ :Noisy Latent          $z_{t-1}$: Denoised Latent

**(b) SphereEdit: Token-Aware Spherical Editing within Diffusion:**

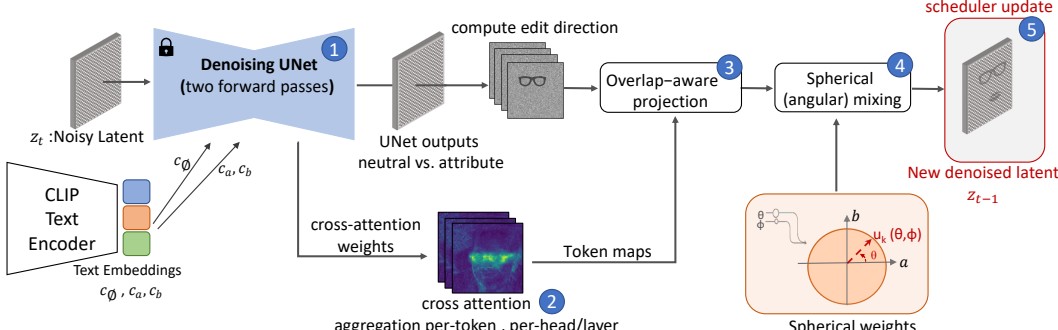

Figure 1: **SphereEdit framework. (a)** Vanilla diffusion: the UNet predicts noise and the scheduler updates the latent from $z_t$ to $z_{t-1}$. **(b)** SphereEdit within the same step: (1) run two UNet passes per attribute (neutral vs. attribute) to form a per-attribute edit direction via noise difference; (2) aggregate cross-attention into token maps—used to detect spatial overlap between attributes; (3) reduce interference in overlap regions via a local masked projection; (4) combine attribute directions with angular (spherical) weights; (5) add the mixed edit to the base prediction and let the scheduler produce $z_{t-1}$. No retraining.

and resolve overlaps between attributes. This geometry + attention formulation enables controllable, composable edits without fine-tuning, while markedly reducing leakage and interference.

In this paper we show that our method SphereEdit yields sharper, more disentangled, and better-localized edits than optimization-, latent-linear, and attention-rewiring baselines, while preserving photorealism and near-standard inference speed. We have three contributions: **(1)** A geometric formulation that models semantic attributes as unit-norm directions on a hypersphere, enabling composable control via angular parameters. **(2)** A training-free, token-aware that uses cross-attention attribution for localized editing. **(3)** An overlap-aware masked orthogonalization that resolves attribute interference only where maps overlap, improving compositionality without retraining.

## 2   Related work

**Diffusion Models and Latent Diffusion.** Diffusion models synthesize images by iteratively denoising latent variables, achieving strong fidelity and diversity [1, 2]. Latent Diffusion (LDM) performs this process in VAE latent space, preserving quality while reducing compute [5]. Our method operates within the standard conditional LDM loop without additional training.

**Image Editing with Diffusion Models.** Training-free editing combines inversion with test-time guidance. DDIM and Null-Text inversion enable faithful reconstruction and improved alignment [2, 15]. Cross-attention methods (Prompt-to-Prompt, MasaCtrl) offer localized control but suffer leakage when composing attributes [12, 16]. Latent-space approaches like DiffusionCLIP use CLIP directions for attribute control [17], while energy-guided methods like SEGA steer sampling via external objectives [18]. Cycle-Diffusion enforces consistency to preserve content [19], and TurboEdit trades precision for speed [20]. Despite these advances, precise multi-attribute composition with locality preservation remains challenging.

**Attention Localization and Compositional Guidance.** Cross-attention aggregation methods like DAAM provide spatial attributions for tokens [21]. Compositional approaches include Composable Diffusion for score combination [22] and SEGA for semantic energies [18]. However, naïve averaging blurs attributes, and concurrent guidance streams interfere in overlapping regions. We use token-aware soft masks and overlap-aware mixing to reduce interference.

**CLIP Semantics and Latent Geometry.** CLIP's joint embedding enables semantic direction control [23], adapted to diffusion in DiffusionCLIP [17]. We leverage hyperspherical geometry where attributes are unit-norm directions with angular composition, combined with attention-derived localization for training-free, disentangled editing.

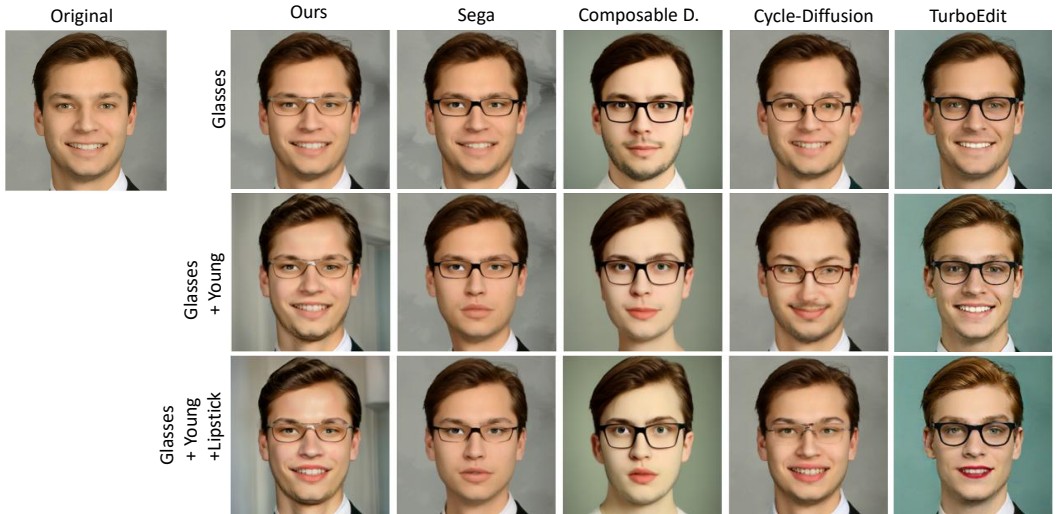

Figure 2: **Qualitative results.** We compare SphereEdit with SEGA [18], Composable Diffusion[22], Cycle Diffusion[19], and TurboEdit[20] for single and composable attributes *Glasses Glasses+Young*, and *Glasses+Young+Lipstick*.

# 3 Method

**SphereEdit** is a training-free framework for composable semantic image editing that combines spherical parameterization with attention-guided map. We first invert the image to the diffusion latent via DDIM inversion [2]; per-attribute directions are computed as noise-prediction differences and mixed with spherical weights. Cross-attention maps are used to detect spatial overlaps between attributes, yielding multi-attribute edits that preserve realism and reduce interference (Fig. 1).

## 3.1 Semantic Direction Derivation

We derive attribute directions in the UNet's noise-prediction space. For attribute $a$ (e.g., smiling), the direction is the difference between conditional and neutral predictions:

$$\mathbf{v}_a = \epsilon_\theta(\mathbf{z}_t, \mathbf{c}_a) - \epsilon_\theta(\mathbf{z}_t, \mathbf{c}_\emptyset), \tag{1}$$

where $\epsilon_\theta$ is the denoiser, $\mathbf{z}_t$ the latent at step $t$, $\mathbf{c}_a$ the attribute text embedding, and $\mathbf{c}_\emptyset$ a neutral/empty embedding. For stable scaling across attributes we use $\hat{\mathbf{v}}_a = \mathbf{v}_a/\|\mathbf{v}_a\|_2$, and bidirectional control.

## 3.2 Spherical Multi-Attribute Composition

We compose attributes with a hyperspherical parameterization that yields intuitive controls. At each step, let $m_a, m_b, m_c \in \{0, 1\}$ indicate which attributes are active. For attributes $(a, b, c)$ we have:

$$\mathbf{u}(\theta, \phi) = w_a\hat{\mathbf{v}}_a + w_b\hat{\mathbf{v}}_b + w_c\hat{\mathbf{v}}_c, \ w_a = s_a\cos\theta\, m_a, \ w_b = s_b\sin\theta\cos\phi\, m_b, \ w_c = s_c\sin\theta\sin\phi\, m_c. \tag{2}$$

where $s_a, s_b, s_c$ are per-attribute gains. This parameterization smoothly interpolates between attributes and keeps coefficients bounded, providing a stable edit combination.

Table 1: **Quantitative results.** We compare SphereEdit with SEGA [18], Composable Diffusion[22], Cycle Diffusion[19], and TurboEdit[20] on *Glasses+Mustache* and *Eyebrow+Lipstick* using CLIP-T[23], CLIP-I[23] and DINO[24] metrics. Higher is better.

| | Glasses + Mustache | | | Eyebrow + Lipstick | | |
|---|---|---|---|---|---|---|
| | Clip-T ↑ | Clip-I ↑ | DINO ↑ | Clip-T ↑ | Clip-I ↑ | DINO ↑ |
| Sega [18] | 0.2226 | 0.7395 | 0.6271 | 0.2230 | 0.6960 | 0.6011 |
| Composable D. [22] | 0.2337 | 0.6353 | 0.3226 | 0.2290 | 0.6227 | 0.3860 |
| Cycle-Diffusion [19] | 0.2260 | 0.6909 | 0.4774 | 0.2238 | 0.6562 | 0.5229 |
| TurboEdit [20] | 0.2203 | 0.6771 | 0.6250 | **0.2333** | 0.5871 | 0.4702 |
| **Ours** | **0.2369** | **0.7573** | **0.7026** | 0.2321 | **0.7275** | **0.6836** |

## 3.3 Attention-Guided Spatial Disentanglement

**Cross-attention maps for overlap detection.** We aggregate cross-attention (`attn2`) across layers/heads into token-aware maps

$$M_a(x,y) = \text{Norm}\left( \sum_\ell \sum_h A^{(\ell,h)}_{(x,y),\text{token}_a} \right), \tag{3}$$

upsampled to $64{\times}64$. These maps provide reliable cues about where different attributes may locate.

**Masked, overlap-aware orthogonalization.** When attributes overlap, mixing causes interference. We detect overlap by thresholding each map. Let $\mathcal{R}_i = \{(x,y) : M_i(x,y) > \tau_i\}$ with $\tau_i = \text{quantile}(M_i, 0.75)$, and $\mathcal{O}_{ij} = \mathcal{R}_i \cap \mathcal{R}_j$. In $\mathcal{O}_{ij}$ we reduce interference via a masked projection:

$$\mathbf{v}_j \leftarrow \mathbf{v}_j - \frac{\langle \mathbf{v}_j, \mathbf{v}_i \rangle_{\mathcal{O}_{ij}}}{\langle \mathbf{v}_i, \mathbf{v}_i \rangle_{\mathcal{O}_{ij}} + \varepsilon}\, \mathbf{v}_i, \quad \langle \mathbf{x}, \mathbf{y} \rangle_{\mathcal{O}_{ij}} = \sum_{(x,y) \in \mathcal{O}_{ij}} \mathbf{x}(x,y)\, \mathbf{y}(x,y). \tag{4}$$

**Edit Integration and Temporal Control.** At step $t$, attributes are active only within user windows $m_i(t) = \mathbf{1}[\, t_{\text{start},i} \le t \le t_{\text{end},i}\,]$, chosen per attribute (e.g., geometry earlier, fine textures later). Let $\tilde{\mathbf{v}}_i$ denote the (optionally orthogonalized) direction from (4). We then apply spherical mixing:

$$\boldsymbol{\epsilon}_{\text{edit}}(t) = \boldsymbol{\epsilon}_{\text{base}}(t) + \sum_i m_i(t)\, w_i\, \tilde{\mathbf{v}}_i. \tag{5}$$

This yields localized, composable edits with reduced interference, while keeping the inference loop unchanged and training-free.

## 4 Experimental Results

Here, we compare SphereEdit against state-of-the-art methods SEGA [18], Composable Diffusion [22], Cycle Diffusion [19], and TurboEdit [20]. For the quantitative result, we use 100 images.

**Qualitative results.** We compare our method with state of the art methods in Figure 2. Our method cleanly applies requested attributes—glasses appear sharp and well-positioned, "young" reduces age-related features without identity distortion, and lipstick is accurately localized. In contrast, SEGA and Cycle-Diffusion often under-edit subtle attributes, Composable Diffusion introduces global changes to tone and structure, and TurboEdit produces unwanted color artifacts. SphereEdit achieves superior locality and composability, preserving identity and minimizing attribute interference even when combining multiple edits.

**Quantitative results.** We compare SphereEdit against state-of-the-art methods in Table 1 on two compositional pairs: Glasses+Mustache and Eyebrow+Lipstick using CLIP-T (text alignment) [23], CLIP-I (identity) [23], and DINO (structure) [24]. Across both pairs, SphereEdit consistently achieves the strongest identity and structural preservation (best CLIP-I/DINO) and delivers top or near-top text alignment (best CLIP-T on Glasses+Mustache, competitive on Eyebrow+Lipstick). These gains reflect SphereEdit's token-aware spatial masking and spherical mixing, which reduce interference between attributes and localize edits without retraining.

# 5   Conclusion

We introduced SphereEdit, a training-free, token-aware editing framework that composes semantic directions on a hypersphere and localizes them with cross-attention. By deriving per-attribute directions from noise differences and mixing them with angular weights—constrained by overlap-aware orthogonalization—SphereEdit achieves precise, composable edits while preserving identity. Quantitatively, it delivers state-of-the-art identity/structure scores and competitive text alignment; qualitatively, it produces sharper, better-localized results than baselines. Current limitations include sensitivity to tokenization and attention quality, and the dependency of time-step selection. Future work will explore per-head/scale selection, and automatic step scheduling.

## Acknowledgments and Disclosure of Funding

This work was undertaken thanks in part to funding from the Connected Minds Program, supported by Canada First Research Excellence Fund, Grant #CFREF-2022-00010.

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

# A  Technical Appendices and Supplementary Material

## A.1  Qualitative results

Figure 3 illustrates how SphereEdit performs training-free, token-aware edits inside the diffusion loop. On the left, we vary a single mixing angle to smoothly traverse between two attributes, demonstrating continuous control without retraining or manual masks. On the right, we show qualitative results animals, and vehicles: for each input we render the two single-attribute edits and their composition, highlighting that edits remain localized (via cross-attention maps) and identity is preserved. Per-attribute directions are computed from noise-prediction differences and combined with spherical weights, which mitigates interference. This figure therefore emphasizes SphereEdit's core properties: smooth angular control, spatial locality, compositionality, and realism.

**SphereEdit: Angular and Compositional Editing**

Figure 3: **SphereEdit.** We introduce a training-free, token-aware diffusion editing method that computes per-attribute directions from noise differences and derives spatial masks from cross-attention. *Left:* Angular control with spherical weights $(w_a, w_b) = (\cos\theta, \sin\theta)$ produces a smooth progression from attribute $a$ ($\theta = 0$) to attribute $b$ ($\theta = \frac{\pi}{2}$), with natural blends at intermediate angles. *Right:* Compositional edits across domains: Input $\to$ A only $\to$ B only $\to$ A+B ($\theta = \frac{\pi}{4}$) for *faces, animals, and vehicles.* S- phereEdit localizes edits with attention-derived masks and composes attributes while preserving identity.

## A.2  Attention maps visualization

**Effect of the attention threshold.**  In figure 3 we vary the gate threshold $\tau$ used to binarize the cross-attention map for a two-attribute edit (glasses + lipstick), holding all other settings fixed. As $\tau$ increases ($0.0 \to 1.0$), the spatial mask becomes sparser: low thresholds admit large regions and cause attribute leakage (skin tone shifts, broader makeup), while moderate values (e.g., $\tau \approx 0.5$) concentrate edits around the rims and lips with fewer collateral changes. Very high thresholds ($\tau \geq 0.8$) over-prune, weakening or fragmenting the lipstick and reducing realism. This illustrates the trade-off between edit strength and locality controlled by $\tau$.

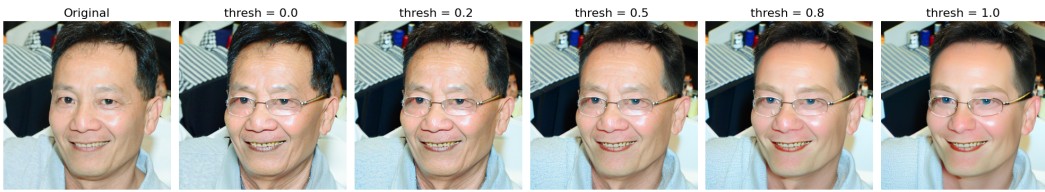

Figure 4: Editing of an image with the attribute *glasses* and *lipstick* at different thresholding value (from left to right: 0, 0.2, 0.5, 0.8, and 1) 0 meaning having an empty mask and 1 meaning no mask apply.

**Effect of token-aware masking (glasses + lipstick)** . Figure 5 shows the attention maps of two attributes ( glasses and lipstick). In the figure, from left to right: we have the original image, the edit without masking, the edit with masking ($\tau = 0.5$), and the cross-attention maps for the *glasses* and *lipstick* tokens. Without masking, the lipstick spills onto skin and global color shifts appear. With masking, attribute directions are gated where the tokens attend, confining glasses to the frames/bridge and lipstick to the mouth while preserving skin tone and identity. The attention maps highlight why: *glasses* has strong, well-localized activations around the eyes, whereas *lipstick* is weaker and more diffuse—explaining why higher $\tau$ can desaturate the lip edit. Overall, token-aware gates reduce leakage and interference in multi-attribute edits.

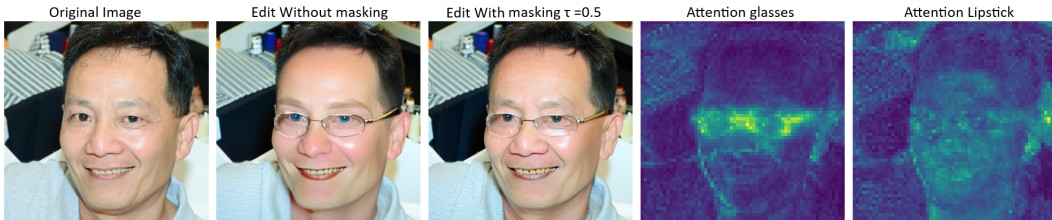

| Original Image | Edit Without masking | Edit With masking τ =0.5 | Attention glasses | Attention Lipstick |

Figure 5: Editing of an image with the attribute *glasses* and *lipstick* with and without thresholding mask and the attention mask for both editing attributes.

