# OpenReview forum: "Reliable Compositional Editing with Overlap-Aware Attention in Diffusion Models"
_NeurIPS.cc/2025/Workshop/Reliable_ML — NeurIPS 2025 - Reliable ML Workshop_

### Official Review · Reviewer_3Sqa · 2025-09-13
**Nice paper, but room to address weakness and more rigorous evals**

**Rating:** 7
**Confidence:** 4

**Review:**

#### SUMMARY
SphereEdit is a training-free framework designed to achieve precise, composable image editing within diffusion models addressing challenges such as slow execution, attribute entanglement, and poor spatial localization in other work.  The core mechanism involves deriving per-attribute directions from noise-prediction differences comparing conditional and neutral predictions and then combining them using angular controls via simple trigonometric operations.
#### STRENGTHS
* No additional training is required.
* Quantitative and qualitative results show improvements or comparable metrics on selective edits compared to other models.
* Masking via token-aware attention allows focused edits ensuring better spatial locatization. This ensures edits are confined to intended areas, preventing, for example, "lipstick altering skin tone"
#### WEAKNESS
* Each new attribute requires a "per-attribute gain" (sa) whose computation is not specified.
* Dependency on user-defined temporal windows:
* In masked overlap-aware orthogonalization, Relying on a static 75th percentile for overlap detection might still allow some subtle interference from regions below this threshold.
* Even though the 75th quantile is dynamically calculated per attribute mask, fixing it at 75th percentile may not be ideal for every different attribute. For example, attribute affecting large image area may work better with lower percentile threshold compared to attribute affecting small image area.
The 75th quantile threshold isn't ideal for attributes affectAttribute Mask's 75th quantile value depends on how much image will be affected by edit which directly affects masked, overlap-aware orthogonalization.
* Upsampling masking to 64x64 may not be suitable for higher resolution images
#### SUGGESTIONS
* Add methodology for computing per-attribute gains
* Clarify the role of θ and ϕ as user controls, not computed parameters:
* u(θ, ϕ) was defined in Spherical multi-attribute composition, paper doesn't explain where and how it is used.
* Test performance on highly correlated attributes (Glasses + Sunglasses).
* Conduct an ablation study to analyze individual component contributions

---

### Official Review · Reviewer_EbMM · 2025-09-20
**Review of SphereEdit**

**Rating:** 6
**Confidence:** 2

**Review:**

The paper introduces SphereEdit, which is a training-free framework for compositional image editing in diffusion models. The proposed method represents semantic attributes as unit-norm directions on the sphere for angular composition. It then employs token-aware attention maps to spatially localize. Experimental results show the superior performance compared to the baselines.

**Strengths**:
- The paper is written well and organized.
- The method is intuitive and straightforward. It is also training-free and can operate in standard diffusion inference loops with minimal overhead.

**Weaknesses**:
- Experiments focus only on facial editing
- The approach seems unable to support general editing
- No ablation study
- The scope of this paper seems a bit misaligned with the workshop topics